# Morphometric differentiation of three chicken ecotypes of Ethiopia using multivariate analysis

**Shishay Markos**[1]*, **Berhanu Belay**[2], **Tadelle Dessie**[3]

**1** Humera Agricultural Research Center of Tigray Agricultural Research Institute, Mekelle, Tigray, Ethiopia, **2** Injibara University, Injibara, Ethiopia, **3** International Livestock Research Institute (ILRI), Addis Ababa, Ethiopia

* shishaymarkos@gmail.com

**Data Availability Statement:** The minimal data that includes mean, minimum and maximum values of the studied quantitative traits are within the paper and its Supporting Information files.

## Abstract

Twenty-one morphometric traits were measured on 770 extensively managed indigenous chickens in the western zone of Tigray, comprising 412 hens and 358 cocks in three agro-ecologies. The quantitative traits for male and female chicken ecotypes were separately analyzed using multivariate analysis with SAS 2008. Four and seven principal components accounted for about 74.26% and 69.77% of the total variability in morphometric traits for males and females, respectively. Earlobe length, wingspan, skull length, and shank length were the most important traits for discriminating among female chicken ecotypes, while wingspan, neck length, earlobe length, spur length, body length, and shank length were the most important discriminatory traits among male chicken ecotypes. The discriminant analysis accurately classified 97.3% of female and 100% of male chicken ecotypes. Cluster analysis revealed the genetic heterogeneity of indigenous chicken populations in both sexes. This finding suggests the presence of morphological variations among the indigenous chicken populations in the different agro-ecological zones, classified as distinct indigenous chicken ecotypes (Lowland, Midland, and Highland). Further DNA-based studies are needed to confirm and complement these morphological variations for effective conservation and the development of sustainable genetic improvement strategies for indigenous chicken populations in the region.

## Introduction

Ethiopia is believed to have the largest livestock population in Africa, consisting of 70.3 million cattle, 42.9 million sheep, 52.5 million goats, 2.1 million horses, 10.8 million donkeys, 0.38 million mules, 8.15 million camels, 56.99 million poultry, and 6.89 million beehives [1]. This sector has made a significant contribution to the country's economic growth [1,2]. Chicken production is an integral part of livestock farming in Ethiopia, where indigenous chickens have played a crucial role in the livelihoods of households, especially those living in poverty. In general, village chickens are seen as a valuable asset due to their easy management and short reproduction cycles [3].

However, the productive performance of indigenous chickens does not match their size, and their low production has obscured their potential to improve the living standards of their

**Funding:** The research project was initially supported through startup funds provided to Mr. Shishay Markos From the Humera Agricultural Research Center of Tigray Agricultural Research Institute with grant number [2130207]. The funder had no role in the study design, data collection, and analysis, decision to publish or preparation of the manuscript.

**Competing interests:** The authors have declared that no competing interests exist.

owners and contribute to the national economy. This might be attributed to poor nutrition, the presence of diseases and predators, institutional and socioeconomic constraints, limited management skills, a poor understanding of the production system, and the absence of comprehensive breeding strategies [3–8]. Efforts to boost the performance of indigenous chickens through crossbreeding with exotic breeds began in 1990 [9]. Unfortunately, this approach was not successful due to the assumption of homogenous village chicken production systems across different parts of Ethiopia. Furthermore, indiscriminate crossbreeding posed a major threat to the indigenous chicken population through breed substitution [4,10].

Baseline information on the diverse gene pool of well-adapted indigenous chickens is crucial for conserving their genetic resources and preserving the genetic variations within and among them. This is necessary to meet current and future market demands and to safeguard against environmental changes, including socio-economic, historical, and cultural shifts. It also ensures sufficient genetic sources for sustainable utilization and improvement. The future sustainable improvement, utilization, and conservation of indigenous chicken genetic resources rely on the genetic variations present within and among them [11].

Several scholars in Ethiopia have conducted phenotypic and genetic characterizations of various indigenous chicken ecotypes. These studies aim to gather valuable information on the genetic variations present within and among the existing chicken genetic resources. This information is crucial for the conservation and utilization of chicken genetic resources. Genetic characterization is the most accurate method for evaluating genetic diversity, but it requires advanced technology and is expensive [12,13]. Researchers also use characterization methods based on morphometric traits that are easy to measure, cost-effective, and provide valuable information [4,14]. It is too complex to analyze and interpret morphological traits using simple univariate and bivariate statistical methods [15]. Furthermore, previous studies by Wu and Lin [16] highlight that biological, biomedical, and agricultural traits are intricate and are influenced by interactions of genetic and non-genetic factors.

Rosario *et al.* [17] also reported that the control mechanisms of morphological traits in chickens are too complex to be explained solely through univariate analysis. All associated traits are biologically correlated due to the pleiotropic effect of genes and loci linkages. Univariate statistical analysis falls short in capturing the collective differences among investigated chicken populations [18]. To address this, multivariate statistical techniques become suitable approaches for concurrently analyzing numerous traits while considering the co-variation that exists between them [19]. Multiple studies have examined the morphological diversity of various indigenous chicken populations in different areas of Ethiopia using multivariate analysis of morphometric traits [20–25]. However, the indigenous chicken populations prevailing in the western zone of Tigray are entirely distinct from the previously studied indigenous chicken populations. This distinction arises from their isolation by physical barriers, differences in agroecology, and significant geographical distance. Furthermore, there has been no research on differentiating these unique indigenous chicken ecotypes found only in the western zone of Tigray regional state of Ethiopia using multivariate analysis of morphometric traits. This study, therefore, aims to bridge this gap by using multivariate analysis of morphometric traits to differentiate between the three indigenous chicken ecotypes in the western zone of the Tigray regional state of Ethiopia.

## Materials and methods

### Ethical statement

To conduct our research, we strictly adhered to Ethiopia's ethical guidelines for accessing genetic resources and community knowledge, as well as benefit sharing [26]. We also followed

the genetic resource transfer and management conventions established by the Directorate Institute of Biodiversity Conservation. Initially, we obtained a supporting letter from the Tigray Agricultural Research Institute, which we presented to the relevant administrative bodies in the study zone. Subsequently, we obtained permission from these bodies to conduct our research in their respective areas. Lastly, we ensured that every farmer who participated in the research provided written informed consent on December 1,2022.

## Description of the study area

The study was conducted in the western zone of the Tigray regional state of Ethiopia, encompassing three agro ecologies: lowland, midland, and highland. These agro ecologies represent three indigenous chicken ecotypes. The lowland chicken ecotype is reared in Kolla (<1500 meters above sea level), the midland in Weynadega (1500–2500 meters above sea level), and the highland in Dega agro-ecology (>2500 meters above sea level). The study zone is located 580–750 km from Mekelle, the capital city of Tigray, with latitude ranging from 13˚ 42' to 14˚ 28' north and longitude from 36˚ 23' to 37˚ 31' east [27]. Annual rainfall in the zone varies from 600 mm to 800 mm, while the annual temperature ranges from 27˚C to 45˚C in Kolla, 15˚C to 30˚C in Weynadega and 10˚C to 22˚C in Dega areas. The zone has an area of 1.5 million hectares and its altitude ranges from 500 to 3008 meters above sea level.

## Sampling techniques

A stratified sampling technique was used to divide the peasant associations of the zone into three categories: lowland (Kolla), midland (Weyna Dega), and highland (Dega). To select the sample peasant associations and farmers (chicken owners), a multi-stage sampling procedure was employed, which included both purposive and random sampling techniques. A total of nine sample peasant associations were purposefully chosen, with four from the lowland, three from the midland, and two from the highland. The selection criteria were based on factors such as village poultry population density, chicken production potential, road accessibility, and representation of the different agro-ecologies in the zone. Similarly, farmers were selected using purposive random sampling, considering their poultry production experience and ownership of three or more chickens.

A total of 770 six-month-old or older chickens (310 from lowland, 260 from midland, and 200 from highland agroecology) were selected by the purposive random sampling technique. The number of chickens per each sample agroecology was determined by proportionate sampling technique based on the size of the chicken populations in the sample agroecologies. Before the main study, a cross-sectional study was conducted to validate the geographical distribution, concentration, and population of the indigenous chicken ecotypes, as well as the peasant associations in each agroecology. This provided a sampling framework from which the sampling units were taken.

## Measurements of quantitative morphometric traits

Morphometric characterization requires at least 10–30 cocks and 100–300 hens [28]. Precision improves as sample sizes increase. Based on this concept, purposive random sampling was used to select 770 indigenous chickens (412 females and 358 males) of both sexes for this study: 310 lowland chickens (146 males and 164 females), 260 midland chickens (120 males and 140 females), and 200 highland chickens (92 males and 108 females). The chickens were about six months old or older, as per the details given by the owners and also confirmed by the researchers using wing plumage. Sixteen characteristics (body weight, body length, skull length, and width, comb length and width, beak length and width, earlobe length and width,

wattle length and width, neck length, wingspan, shank length, and spur length) were measured based on the methodology developed by FAO[28] and Francesch *et al.*[29], and five zoometric traits (skull index, comb index, earlobe index, beak index, and wattle index) were measured based on the methodology developed by Francesch *et al.* [29].

The sampled chickens' wingspan, neck, body, and shank length were measured using a measuring tape (+1mm), and their live body weight was measured using a sensitive balance with an electronic weighing scale (precision = 1 g). The comb length, comb width, earlobe length, earlobe width, wattle length, wattle width, skull length, skull width, beak length, beak width, and spur length of the chickens were measured using a caliper (+0.01mm). The same individual took each measurement early in the morning before the chickens were fed. Five corporals' indexes were derived from the above-evaluated characteristics and determined following the methodology developed by Francesch *et al.* [29]. They express the relation between the length and width of the structure or respective quantitative traits.

$$\text{Skull index} = \frac{\text{Skull length}}{\text{Skull width}} \qquad \text{Comb index} = \frac{\text{Comb length}}{\text{Comb width}}$$

$$\textbf{Earlobe index} = \frac{\text{Earlobe length}}{\text{Earlobe width}} \qquad \text{Wattle index} = \frac{\text{Wattle length}}{\text{Wattle length}}$$

$$\text{Beak index} = \frac{\text{Beak length}}{\text{Beak width}}$$

## Data analysis

**Univariate analysis.** The General linear model procedure (PROC GLM) of SAS version 9.2 [30] was employed to determine the effects of chicken sex and agroecology (chicken ecotypes) on the measured quantitative morphological traits. Significant means were separated using the Tukey test. The statistical model used was:

$$\mathbf{Y_{ijk} = \mu + E_i + S_j + ES_{ij} + E_{ijk}} \tag{1}$$

Where $\mathbf{Y_{ijk}}$: The corresponding quantitative traits in the i$^{th}$ chicken ecotype (i = 3, lowland, midland and highland)

$\mu$ = overall population mean for corresponding quantitative trait

$E_i$ = the fixed effect of i$^{th}$ chicken ecotype (i = 3. Lowland, midland and highland)

$S_j$ = the fixed effect of the j$^{th}$ chicken sexes (i = male and female)

$ES_{ij}$ = the interaction of chicken ecotype and sex and $E_{ijk}$ = residual error

**Multivariate analysis.** principal component analysis (PCA), canonical discriminant analysis (CDA), stepwise discriminant analysis, and cluster analysis, was performed using Statistical Analysis System (SAS) version 9.2 [30] separately for each sex. Mature indigenous chickens' measurable morphological characters were utilized for conducting the principal component analysis. This method facilitates the transformation of a large number of variables into a smaller set of uncorrelated latent variables, known as principal components (PCs). Cluster and principal component analyses were computed by using the procedures CLUSTER and PRINCOMP, respectively, using SAS software version 9.2 [30]. With the help of a Dendogram, the average linkage cluster method was used to group morphological similarity or divergence of three indigenous chicken ecotypes. The stepwise discriminate procedure of SAS was applied using PROCSTEPDISC to determine which morphological traits have more discriminating power than others in discriminating the genetic groups. The relative value of the morphometric variables in differentiating among the three chicken populations was determined by

evaluating the significance level, F statistics, and partial $R^2$. The canonical discriminant analysis was conducted through the CANDISC procedure of SAS, enabling both univariate and multivariate one-way analyses to calculate the Mahalanobis distance among the chicken populations [14,31,32]. The ability of these canonical functions to correctly assign each animal to its respective group was assessed as the percentage of accurate assignments using the DISCRIM procedure [31,33,34].

## Results

The overall mean±SE for all considered quantitative traits of the three chicken ecotypes is presented in Table 1. The results indicate that the sex-by-chicken ecotype interaction significantly (P<0.05) affects all considered quantitative traits (Table 1). Lowland male chicken ecotypes displayed significantly (P<0.05) higher values of body weight, body length, shank length, comb length, beak length, and wingspan compared to midland males, while highland male chicken ecotypes exhibited the lowest values. Similarly, lowland male chicken ecotypes had higher mean values of comb width, earlobe length, earlobe width, spur length, wattle length, and wattle width, followed by highland males, with midland males recording the lowest values. In contrast, midland male chicken ecotypes showed significantly (P<0.05) higher values of comb index and neck length compared to lowland males, while highland males had the lowest values. The mean values of earlobe index, beak index, and skull length for both lowland and highland male chicken ecotypes were not significantly different but were significantly lower than those of midland males. Both lowland and midland male chicken ecotypes exhibited similar skull index values, which were significantly longer than those of highland males. Midland and highland male chicken ecotypes had the same beak width, which was significantly lower than that of lowland males. There were no significant differences among the three male chicken ecotypes in terms of wattle and skull indices.

Similarly, significantly longer body length and wingspan were observed in lowland female chicken ecotypes, followed by midland females, while highland female chicken ecotypes displayed the lowest values. Both lowland and midland female chicken ecotypes had the same body weight and earlobe index, which were significantly higher than those of highland females. However, midland and highland female chicken ecotypes exhibited similar shank length, which was significantly lower than that of lowland females. The mean values of comb length, comb index, wattle width, and beak width for both lowland and highland female chicken ecotypes were not significantly different but were significantly longer than those of midland females. Highland female chicken ecotypes had the highest earlobe length and width, followed by lowland females, while midland females recorded the lowest values. Both lowland and highland female chicken ecotypes exhibited the same wattle and beak indices, which were significantly lower than those of midland females. Midland and highland female chicken ecotypes had the same skull length, which was significantly higher than that of lowland females. All three female chicken ecotypes had the same values of comb width, wattle length, skull width, skull index, neck length, beak length, and spur length. The additional information for quantitative traits for female chicken ecotypes is provided in (S1 Table), and for male chicken ecotypes in (S2 Table).

### Principal component analysis (PCA)

The PCA revealed that four PCs were extracted and retained for males, while seven PCs were retained for females, based on the Eigen value-one criterion (Kaiser Criterion) which states that any component with an Eigen value larger than one should be retained and interpreted [24]. The first PC explains approximately 42.4% (Eigen value = 8.90), PC2 contributes 16.97%

**Table 1. Least square means for quantitative traits of indigenous chicken ecotypes in three agro-ecological zones of western zone of Tigray (Lsmeans±SEM).**

| Traits | | Agro-ecological zones | | | |
|---|---|---|---|---|---|
| | Chicken sex | Lowland (N = 310) | Midland (N = 260) | Highland (N = 200) | Total |
| Body length (cm) | Male | 39.53 ± 0.09[a] | 36.08 ± 0.10[b] | 32.50± 0.12[c] | 36.04±0.06 [a] |
| | Female | 24.23 ±0.09[e] | 24.92±0.09[d] | 23.82±0.11 [f] | 24.32±0.06 [b] |
| Body wt(gm) | Male | 1.676±0.01[a] | 1.579±0.01[b] | 1.451±0.01[c] | 1.569±0.004[a] |
| | Female | 1.272±0.01[d] | 1.270±0.01[d] | 1.192±0.01[e] | 1.261±0.004[b] |
| Shank length (cm) | Male | 12.93 ±0.05[a] | 10.81±0.06[b] | 9.87±0.06[c] | 11.20± 0.03[a] |
| | Female | 8.81±0.05[c] | 8.29±0.05[e] | 8.27 ±0.06[e] | 8.46± 0.03[b] |
| Comb length (cm) | Male | 7.50 ±0.04[a] | 6.68±0.04[b] | 5.49±0.05[c] | 6.55±0.02[a] |
| | Female | 2.73±0.03[d] | 2.57±0.04[e] | 2.78 ±0.04[d] | 2.69± 0.02[b] |
| Comb width(cm) | Male | 3.81 ±0.02[a] | 2.75 ±0.02[c] | 3.09±0.02[b] | 3.22±0.01[a] |
| | Female | 1.47 ±0.02[d] | 1.51±0.02[d] | 1.47 ±0.02[d] | 1.48±0.01[b] |
| Comb index | Male | 1.98 ±0.02[b] | 2.44 ±0.02[a] | 1.78 ±0.03[de] | 2.07± 0.01[a] |
| | Female | 1.87±0.02[cd] | 1.71 ±0.02[e] | 1.89±0.02[bc] | 1.83±0.01[b] |
| Earlobe length (cm) | Male | 3.55 ±0.01[a] | 2.53 ±0.02[c] | 3.06 ±0.02[b] | 3.05±0.01[a] |
| | Female | 1.63±0.01 [e] | 1.27±0.01[f] | 1.88±0.02[d] | 1.59±0.01[b] |
| Earlobe width(cm) | Male | 2.19±0.02 [a] | 1.32 ±0.02[c] | 1.86 ±0.02[b] | 1.79± 0.01[a] |
| | Female | 0.91 ±0.01[e] | 0.71 ±0.02[f] | 1.23 ±0.02 [d] | 0.95±0.01[b] |
| Earlobe index | Male | 1.64±0.02[c] | 1.96 ±0.03[a] | 1.66 ±0.02[c] | 1.75± 0.01[a] |
| | Female | 1.84 ±0.02[b] | 1.8 4 ±0.02[b] | 1.57 ±0.03[c] | 1.75±0.01[a] |
| Wattle length (cm) | Male | 6.21± 0.05[a] | 4.46 ±0.06[c] | 5.30 ±0.06[b] | 5.32±0.03[a] |
| | Female | 2.12 ±0.05[d] | 2.02 ±0.05[d] | 2.08± 0.06[d] | 2.07 ±0.03[b] |
| Wattle width(cm) | Male | 3.93± 0.04[a] | 2.77 ±0.04[c] | 3.37 ±0.05[b] | 3.36±0.03[a] |
| | Female | 1.38 ±0.04[d] | 1.08 ±0.04[e] | 1.39 ±0.05[d] | 1.28±0.02[b] |
| Wattle index | Male | 1.61±0.02[bc] | 1.71 ±0.03[b] | 1.66 ±0.04[b] | 1.66±0.02[b] |
| | Female | 1.61± 0.03[bc] | 2.02 ±0.03[a] | 1.52 ±0.05[c] | 1.71±0.02[a] |
| Skull length(cm) | Male | 6.63 ±0.07[b] | 7.04 ±0.08[a] | 6.63± 0.09[b] | 6.77±0.04[a] |
| | Female | 5.78 ±0.06[d] | 6.27 ±0.07[c] | 6.38 ±0.08[bc] | 6.15± 0.04[b] |
| Skull width(cm) | Male | 4.18 ±0.05[a] | 4.18 ±0.06 [a] | 3.87 ±0.07[b] | 4.08±0.03[a] |
| | Female | 3.41 ±0.05[c] | 3.51 ±0.05[c] | 3.59 ±0.06[c] | 3.50±0.03[b] |
| Skull index | Male | 1.64 ±0.02 [c] | 1.72 ±0.02[bc] | 1.73 ±0.03[abc] | 1.70±0.01[b] |
| | Female | 1.76 ± 0.02[ab] | 1.82±0.02 [a] | 1.80 ±0.02[ab] | 1.8±0.01[a] |
| Neck length(cm) | Male | 15.70 ± 0.07[b] | 16.93 ±0.08[a] | 13.70± 0.09[c] | 15.44± 0.05[a] |
| | Female | 13.73±0.07[c] | 13.61±0.08[c] | 12.15 ±0.09[c] | 13.16± 0.05[b] |
| Beak length(cm) | Male | 2.23± 0.02[a] | 2.07 ±0.02[b] | 1.99 ±0.02[c] | 2.10±0.01[a] |
| | Female | 2.03 ±0.02[bc] | 2.06±0.02[b] | 2.06 ±0.02[bc] | 2.05±0.01[b] |
| Beak width(cm) | Male | 1.24± 0.01[a] | 0.97 ±0.02[cd] | 1.04 ±0.02[bc] | 1.08±0.10[a] |
| | Female | 1.08 ± 0.01[b] | 0.93 ±0.02[d] | 1.03 ±0.02[bc] | 1.01±0.01[b] |
| Beak index | Male | 1.80± 0.03[d] | 2.17 ± 0.03[b] | 1.93 ±0.04[cd] | 1.97±0.02[b] |
| | Female | 1.97 ±0.03[c] | 2.35± 0.03[a] | 2.01 ±0.03[c] | 2.11±0.02 [a] |
| Spur length(cm) | Male | 2.44 ±0.01[a] | 1.48 ±0.01[c] | 1.87 ±0.02[b] | 1.93± 0.01[a] |
| | Female | 0.42 ±0.01[d] | 0.37 ±0.01[d] | 0.41 ± 0.01[d] | 0.40 ± 0.01[b] |
| Wing span(cm) | Male | 47.52 ±0.14[a] | 40.01 ±0.15[b] | 36.80 ±0.18[c] | 41.44± 0.09[a] |
| | Female | 36.05±0.13[d] | 34.39±0.14[e] | 32.83±0.16[f] | 34.42±0.08[b] |

Ls means with different superscripts are significantly different (p<0.05).

**Table 2. Eigen vectors and Eigen values of the four retained principal components for the 21 quantitative traits of indigenous male chicken ecotypes in western Tigray.**

| Traits | Eigen vectors | | | | Communality |
|---|---|---|---|---|---|
| | PC1 | PC2 | PC3 | PC4 | |
| Body length | 0.24 | 0.28 | 0.15 | 0.04 | 0.16 |
| Body weight | 0.22 | 0.27 | -0.01 | -0.10 | 0.14 |
| Shank length | 0.28 | 0.18 | 0.07 | -0.13 | 0.13 |
| Comb length | 0.22 | 0.30 | 0.14 | -0.07 | 0.17 |
| Comb width | 0.30 | -0.09 | 0.01 | -0.03 | 0.10 |
| Comb index | -0.12 | 0.39 | 0.13 | -0.04 | 0.19 |
| Earlobe length | 0.27 | -0.17 | 0.08 | 0.12 | 0.12 |
| Earlobe width | 0.27 | -0.23 | -0.01 | -0.10 | 0.14 |
| Earlobe index | -0.16 | 0.23 | 0.11 | 0.28 | 0.17 |
| Wattle length | 0.28 | -0.10 | -0.11 | -0.15 | 0.12 |
| Wattle width | 0.26 | -0.09 | -0.24 | -0.23 | 0.19 |
| Wattle index | -0.06 | 0.09 | -0.05 | 0.63 | 0.40 |
| Skull length | 0.03 | 0.19 | -0.52 | -0.05 | 0.31 |
| Skull width | 0.10 | 0.19 | -0.59 | 0.06 | 0.40 |
| Skull index | -0.09 | -0.09 | 0.40 | -0.26 | 0.24 |
| Neck length | -0.01 | 0.46 | 0.07 | -0.02 | 0.22 |
| Beak length | 0.22 | 0.17 | 0.17 | 0.06 | 0.11 |
| Beak width | 0.26 | -0.06 | 0.09 | 0.36 | 0.21 |
| Beak index | -0.20 | 0.18 | -0.01 | -0.41 | 0.24 |
| Spur length | 0.29 | -0.13 | 0.09 | 0.10 | 0.12 |
| Wing span | 0.29 | 0.18 | 0.10 | 0.01 | 0.13 |
| Eigen value | 8.90 | 3.56 | 2.03 | 1.10 | |
| Difference | 5.34 | 1.54 | 0.92 | | |
| % of total variance | 42.4 | 16.97 | 9.64 | 5.24 | |
| Cumulative (%) | 42.4 | 59.37 | 69.01 | 74.26 | |

(Eigen value = 3.56), PC3 accounts for 9.64% (Eigen value = 2.03), and PC4 accounts for 5.24% (Eigen value = 1.10) of the total variations in the 21 quantitative traits of the three male chicken ecotypes. These retained PCs provide a comprehensive summary of the data, explaining about 74.26% of the total morphometric trait variability in the male chicken ecotypes (Table 2 and **Fig 1**).

The analysis of the Eigen Vectors (loadings) of the PCs indicates that PC1 has high loadings on shank length, comb width, earlobe length, earlobe width, wattle length, wattle width, beak length, spur length, and wingspan. Therefore, PC1 can be considered a measure of the variations in these traits among the three male chickens. The variables most associated with PC2 are body length, body weight, comb length, comb index, and neck length, and this factor is mainly responsible for the variability of these traits among the male chickens. PC3 shows a higher loading on skull length, skull width, and skull index, and is primarily a measure of the variability of skull traits. On the other hand, PC4 is highly correlated with earlobe index, wattle index, beak length, and beak index, serving as a central measure of the variability of these traits among the male chicken ecotypes.

In the female chicken ecotypes, PC1 accounted for 20.01% of the observed variations (Eigen value = 4.20). It had higher loadings mainly on earlobe width, earlobe length, wattle width, and wattle index. PC2 explained approximately 12.69% of the observed morphometric variation (Eigen value = 2.67) and had higher loadings on skull length, skull width, skull index,

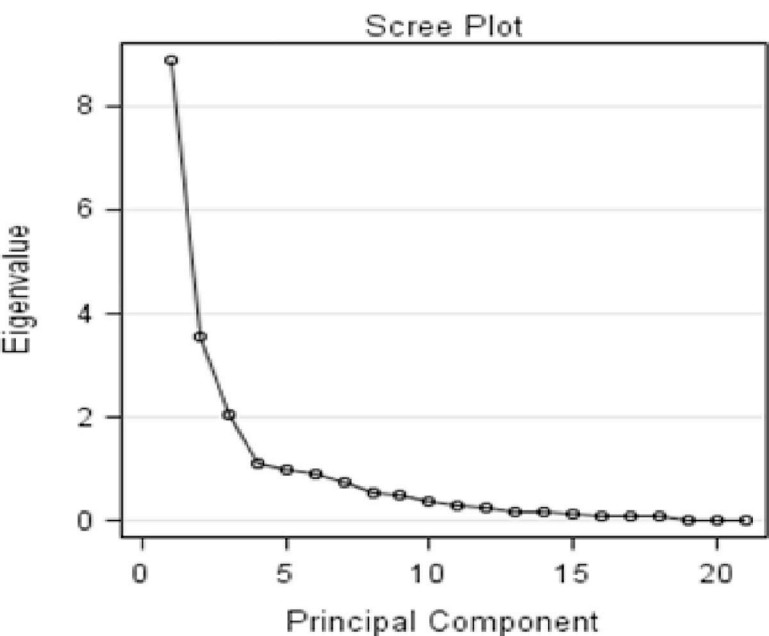

**Fig 1. Scree plot of eigen value to component number of male local chicken ecotypes.**

and wingspan, which primarily expressed the variations of these traits among female ecotypes. Moreover, PC3 accounted for about 9.23% of the total variation (Eigen value = 1.94), mainly on body weight and wattle length. PC4 also explained 8.94% of the observed variation in morphometric traits (Eigen value = 1.88), primarily related to the comb index, beak index, comb length, and beak width. PC5 contributed to 8.36% of the total observed morphometric variations (Eigen value = 1.76), mainly associated with comb width and beak index. PC6 explained about 5.47% of the total variation (Eigen value = 1.15), mostly related to body length and comb length, while PC7 accounted for 5.05% of the total variability (Eigen value = 1.106) in the quantitative traits, chiefly on shank length, wattle length, beak length, and spur length of the female chicken ecotypes. Overall, the seven retained principal components provide a comprehensive summary of the data, explaining about 89.77% of the variability in the 21 morphometric traits of the female chicken ecotypes (Table 3 and **Fig 2**).

Communality refers to the total sum of the squared loadings of a given variable. It represents the proportion of variation in that variable that is explained by the retained principal components. The results suggest that the principal component analysis is most effective in explaining the variation in wattle index, skull width, skull index, beak index, neck length, and beak width of the male chicken ecotypes (Table 2).

Similarly, the principal component analysis of female indigenous chicken ecotypes indicated that the seven retained principal components explained 52%, 49%, 47%, 47%, 43%, 41%, 36%, and 35% of the variations in comb index, beak length, beak width, beak index, comb length, wattle length, skull width, and wattle width, respectively (Table 3).

## Stepwise discriminate analysis

For females, the stepwise discriminant analysis revealed that sixteen variables (earlobe length, wingspan, skull length, shank length, earlobe width, neck length, body length, beak index, beak length, wattle index, body weight, earlobe index, comb index, wattle length, wattle width, and skull index) show potential discriminatory power in distinguishing the three female

**Table 3. Eigenvectors and Eigen values of the Correlation Matrix for 21 morphometric traits of indigenous female chicken ecotypes in western Tigray.**

| Traits | Eigen vectors | | | | | | | Communality |
|---|---|---|---|---|---|---|---|---|
| | PC1 | PC2 | PC3 | PC4 | PC5 | PC 6 | PC 7 | |
| Body Length | -0.12 | 0.27 | -0.05 | 0.14 | 0.09 | 0.41 | -0.11 | 0.30 |
| Body weight | 0.03 | 0.08 | 0.39 | 0.09 | 0.04 | -0.13 | 0.19 | 0.22 |
| Shank length | 0.16 | 0.21 | 0.26 | -0.16 | -0.01 | -0.15 | 0.35 | 0.31 |
| Comb length | 0.24 | -0.12 | 0.30 | 0.35 | -0.06 | 0.38 | 0.06 | 0.43 |
| Comb width | -0.01 | -0.03 | 0.21 | -0.23 | 0.45 | 0.04 | -0.19 | 0.34 |
| Comb index | 0.22 | -0.11 | 0.16 | 0.43 | -0.33 | 0.32 | 0.17 | 0.52 |
| Earlobe length | 0.32 | -0.21 | -0.12 | -0.01 | -0.18 | -0.27 | -0.12 | 0.28 |
| Earlobe width | 0.37 | -0.25 | -0.17 | 0.10 | 0.07 | -0.19 | 0.01 | 0.28 |
| Earlobe index | -0.24 | 0.15 | 0.13 | -0.15 | -0.27 | 0.025 | -0.14 | 0.21 |
| Wattle length | 0.25 | 0.12 | 0.36 | -0.05 | 0.24 | 0.09 | -0.37 | 0.41 |
| Wattle width | 0.36 | 0.13 | 0.26 | -0.04 | 0.13 | -0.06 | -0.34 | 0.35 |
| Wattle index | -0.32 | -0.07 | 0.04 | 0.02 | 0.10 | 0.20 | 0.11 | 0.17 |
| Skull length | 0.15 | 0.33 | -0.32 | 0.11 | 0.20 | 0.11 | 0.04 | 0.30 |
| Skull width | 0.17 | 0.44 | -0.29 | 0.21 | 0.05 | -0.04 | -0.07 | 0.36 |
| Skull index | -0.10 | -0.34 | 0.17 | -0.22 | 0.17 | 0.19 | 0.18 | 0.30 |
| Neck length | -0.21 | 0.22 | 0.27 | 0.03 | -0.21 | -0.11 | -0.05 | 0.23 |
| Beak length | 0.20 | 0.17 | -0.09 | -0.21 | 0.28 | 0.21 | 0.50 | 0.49 |
| Beak width | 0.24 | 0.08 | -0.10 | -0.48 | -0.22 | 0.33 | 0.08 | 0.47 |
| Beak index | -0.18 | 0.04 | 0.08 | 0.40 | 0.40 | -0.25 | 0.22 | 0.47 |
| Spur length | 0.17 | 0.07 | 0.13 | -0.10 | -0.10 | -0.30 | 0.34 | 0.28 |
| Wing span | -0.04 | 0.40 | 0.17 | -0.04 | -0.25 | -0.10 | 0.07 | 0.27 |
| Eigen value | 4.20 | 2.67 | 1.94 | 1.88 | 1.76 | 1.15 | 1.06 | |
| Difference | 1.54 | 0.73 | 0.06 | 0.12 | 0.61 | 0.09 | | |
| %of total variance | 20.01 | 12.69 | 9.23 | 8.94 | 8.36 | 5.47 | 5.05 | |
| Cumulative (%) | 20.01 | 32.7 | 41.93 | 50.88 | 59.24 | 64.71 | 69.77 | |

chicken ecotypes (Table 4). By comparing the F-value, partial $R^2$-value, and p-value statistics of each significant explanatory variable, it was found that earlobe length exhibited the most significant discriminative strength in differentiating the three female chicken ecotypes, while skull breadth had the least significant discriminative ability.

Similarly, the stepwise discriminant analysis identified 16 variables (wingspan, neck length, earlobe length, spur length, body length, skull length, shank length, earlobe index, comb length, wattle length, comb index, beak width, body weight, beak index, wattle index, and wattle width) out of the 21 quantitative traits that showed potential discriminatory power in distinguishing the three male chicken ecotypes. Based on the F-value, partial $R^2$-value, and p-value statistics for each significant explanatory variable, wingspan exhibited the most significant discriminative potential, while wattle width displayed the least significant discriminative power in differentiating the three male chicken ecotypes (Table 5).

## Canonical discriminate analysis

The canonical discriminant analysis revealed the extraction of two canonical variables for female chicken ecotypes, which accounted for 100% of the total variations (Table 6). The first canonical variable (CAN1), also known as the Fisher linear discriminant function, explained 63.58% of the total variation, which can be considered reasonable. Additionally, the second canonical variable (CAN2) explained 36.42% of the total variations among female chicken ecotypes.

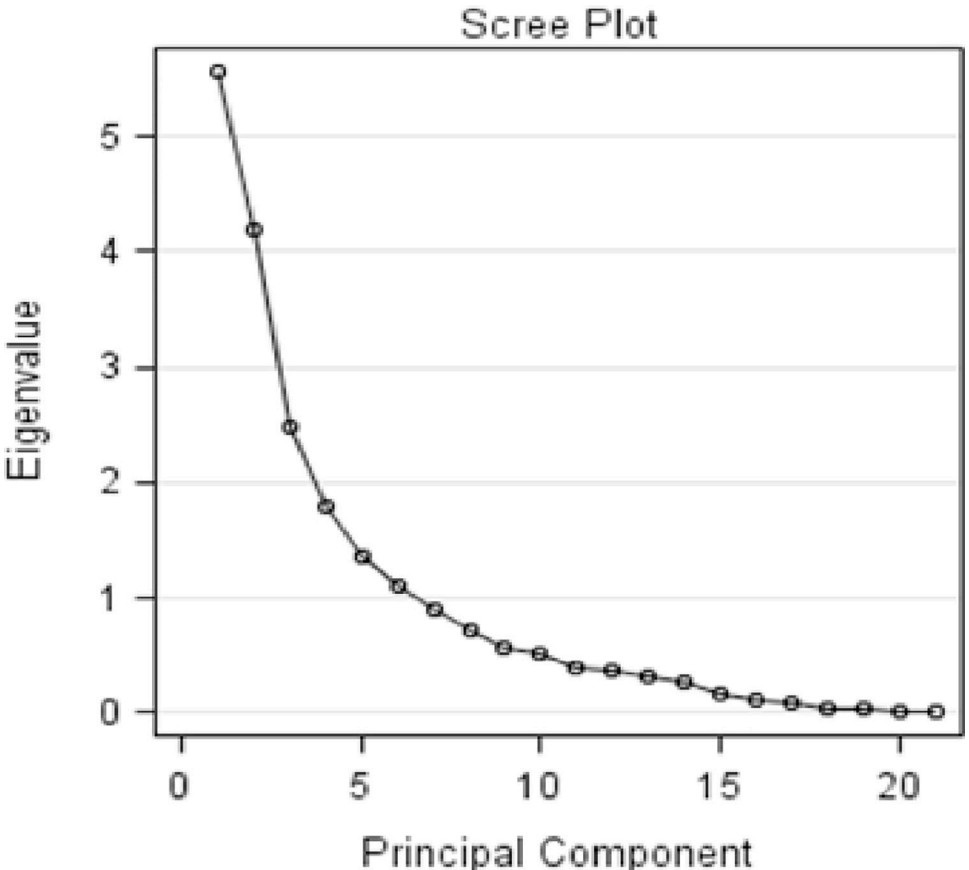

**Fig 2. Scree plot of eigen value to component number of female local chicken ecotypes.**

Similar to the female chicken ecotypes, two canonical variates were extracted from the canonical discriminant analysis for male chicken ecotypes (Table 6). The first canonical variable accounted for 70.06% of the total variations, while the second canonical variable (CAN2) accounted for 29.94% of the total variations among the male chickens. Together, the two canonical variables accounted for 100% of the total morphometric variations among the male chicken ecotypes.

The standardized total-sample canonical coefficients obtained from the canonical discriminant analysis provided insights into how each original variable aligned with each of the two canonical variables, assigning weights to each trait based on their contribution to the formation of the extracted canonical variables.

In the case of female chicken ecotypes, body length, body weight, earlobe length, earlobe width, earlobe index, skull width, and beak length had relatively higher weights in the extraction of CAN1. On the other hand, the explanatory variables shank length, comb length, comb width, comb index, wattle length, wattle width, wattle index, skull length, skull index, neck length, beak width, beak index, spur length, and wingspan made significant contributions, in that order, to the second canonical variable (CAN2). CAN1 exhibited higher discriminatory power compared to CAN2, as the CAN1 axis displayed greater differentiation and distribution of variates among the female ecotypes. This indicates that earlobe width, earlobe length, earlobe index, body length, beak length, skull width, and body weight carried more weight in CAN1 and can be considered the most influential variables for differentiating among the female chicken ecotypes.

**Table 4. Summary of step wise selection of traits through the STEPDISC Procedure for female chicken ecotype.**

| Step | Traits | Partial R2 | F Value | Pr> F | Wilks'ʎ | Pr<ʎ | ASCC | Pr>ASCC |
|---|---|---|---|---|---|---|---|---|
| 1 | Earlobe Length | 0.731 | 554.93 | < .0001 | 0.26928 | < .0001 | 0.36536 | < .0001 |
| 2 | Wing Span | 0.408 | 140.31 | < .0001 | 0.15955 | < .0001 | 0.56632 | < .0001 |
| 3 | Skull Length | 0.279 | 78.90 | < .0001 | 0.11497 | < .0001 | 0.65049 | < .0001 |
| 4 | Shank Length | 0.210 | 53.86 | < .0001 | 0.09086 | < .0001 | 0.69596 | < .0001 |
| 5 | Earlobe Width | 0.167 | 40.64 | < .0001 | 0.07568 | < .0001 | 0.71864 | < .0001 |
| 6 | Neck Length | 0.055 | 11.71 | < .0001 | 0.07153 | < .0001 | 0.72691 | < .0001 |
| 7 | Body Length | 0.049 | 10.45 | < .0001 | 0.068 | < .0001 | 0.73274 | < .0001 |
| 8 | Beak Index | 0.045 | 9.52 | < .0001 | 0.06493 | < .0001 | 0.73981 | < .0001 |
| 9 | Beak Length | 0.043 | 8.99 | 0.0002 | 0.06214 | < .0001 | 0.74441 | < .0001 |
| 10 | Wattle Index | 0.034 | 7.03 | 0.001 | 0.06003 | < .0001 | 0.74909 | < .0001 |
| 11 | Body Weight | 0.029 | 6.04 | 0.0026 | 0.05827 | < .0001 | 0.7522 | < .0001 |
| 12 | Earlobe Index | 0.024 | 4.98 | 0.0073 | 0.05684 | < .0001 | 0.75464 | < .0001 |
| 13 | Comb Index | 0.018 | 3.62 | 0.0276 | 0.05583 | < .0001 | 0.75726 | < .0001 |
| 14 | Wattle Length | 0.013 | 2.66 | 0.071 | 0.05508 | < .0001 | 0.7586 | < .0001 |
| 15 | Wattle Width | 0.024 | 4.83 | 0.0084 | 0.05377 | < .0001 | 0.76196 | < .0001 |
| 16 | Skull Width | 0.012 | 2.36 | 0.0961 | 0.05329 | < .0001 | 0.76275 | < .0001 |

All the variables in the table above are found to have potential discriminatory power. These variables are used to develop discrimination models in both the CANDISC and DISCRIM procedure.

For male chicken ecotypes, body length, shank length, comb length, earlobe width, earlobe index, wattle width, skull length, skull index, beak length, beak width, spur length, and wing-span had relatively higher weights in the extraction of CAN1. Conversely, body weight, comb width, comb index, earlobe length, wattle length, wattle index, skull width, neck length, and beak index made significant contributions, in that order, to the second canonical variable

**Table 5. Summary of step wise selection of traits through the STEPDISC Procedure for male chicken ecotypes.**

| Step | Traits | Partial R2 | F Value | Pr> F | Wilks'ʎ | Pr<ʎ | ASCC | Pr>ASCC |
|---|---|---|---|---|---|---|---|---|
| 1 | Wing span | 0.8574 | 1067.53 | < .0001 | 0.14257 | < .0001 | 0.42872 | < .0001 |
| 2 | Neck length | 0.6994 | 411.9 | < .0001 | 0.04285 | < .0001 | 0.77766 | < .0001 |
| 3 | Earlobe length | 0.4717 | 157.61 | < .0001 | 0.02264 | < .0001 | 0.84312 | < .0001 |
| 4 | Spur length | 0.2626 | 62.69 | < .0001 | 0.01669 | < .0001 | 0.86148 | < .0001 |
| 5 | Body length | 0.2342 | 53.69 | < .0001 | 0.01278 | < .0001 | 0.88086 | < .0001 |
| 6 | Skull length | 0.2341 | 53.49 | < .0001 | 0.00979 | < .0001 | 0.89261 | < .0001 |
| 7 | Shank length | 0.164 | 34.22 | < .0001 | 0.00818 | < .0001 | 0.89868 | < .0001 |
| 8 | Earlobe index | 0.1398 | 28.28 | < .0001 | 0.00704 | < .0001 | 0.90754 | < .0001 |
| 9 | Comb length | 0.1318 | 26.34 | < .0001 | 0.00611 | < .0001 | 0.9138 | < .0001 |
| 10 | Wattle length | 0.1292 | 25.66 | < .0001 | 0.00532 | < .0001 | 0.92162 | < .0001 |
| 11 | comb index | 0.0742 | 13.83 | < .0001 | 0.00493 | < .0001 | 0.92369 | < .0001 |
| 12 | Beak width | 0.0642 | 11.79 | < .0001 | 0.00461 | < .0001 | 0.92603 | < .0001 |
| 13 | Body weight | 0.0643 | 11.79 | < .0001 | 0.00431 | < .0001 | 0.92935 | < .0001 |
| 14 | Beak index | 0.0532 | 9.62 | < .0001 | 0.00409 | < .0001 | 0.93094 | < .0001 |
| 15 | Wattle index | 0.0299 | 5.25 | 0.0057 | 0.00396 | < .0001 | 0.93213 | < .0001 |
| 16 | Wattle width | 0.0244 | 4.25 | 0.015 | 0.00387 | < .0001 | 0.93282 | < .0001 |

ASCC = Average Squared Canonical Correlation. All the variables in the above table are found to have potential discriminatory power. These variables are used to develop discrimination models in both the CANDISC and DISCRIM procedure

**Table 6. Total–sample standardized Canonical Coefficients, Canonical correlations, class means on canonical variables and total variation explained by each variate of the indigenous chicken ecotypes.**

| Canonical correlations and total variation explained by each | | | | |
|---|---|---|---|---|
| Variance explained | **Female** chicken ecotypes | | **Male** chicken ecotypes | |
| | CAN 1 | CAN2 | CAN 1 | CAN2 |
| Variance (no) | 4.52 | 2.59 | 23.26 | 9.94 |
| Variance (%) | 63.58 | 36.42 | 70.06 | 29.94 |
| F value | 63.86 | 50.44 | 243.91 | 166.97 |
| Pr>F | < .0001 | < .0001 | < .0001 | < .0001 |
| Adjusted Canonical correlation | 0.905 | 0.849 | 0.979 | 0.953 |
| Total–sample standardized Canonical Coefficients | | | | |
| Traits | **Female** chicken ecotypes | | **Male** chicken ecotypes | |
| | CAN 1 | CAN2 | CAN 1 | CAN2 |
| Body length | -0.30 | -0.06 | 1.18 | -0.68 |
| Body weight | -0.17 | -0.07 | -0.03 | -0.48 |
| Shank length | 0.09 | 0.66 | 0.62 | -0.22 |
| Comb length | -0.19 | -2.11 | 0.77 | -0.09 |
| Comb width | 0.06 | 1.38 | -0.07 | -0.33 |
| Comb index | 0.13 | 2.48 | -0.49 | -0.73 |
| Earlobe length | 0.79 | 0.21 | 0.54 | 0.79 |
| Earlobe width | 1.60 | -0.01 | 0.50 | 0.46 |
| Earlobe index | 0.56 | 0.09 | 0.18 | -0.07 |
| Wattle length | -0.35 | -0.69 | -0.13 | 0.56 |
| Wattle width | 0.29 | 0.84 | 0.26 | 0.26 |
| Wattle index | -0.03 | 0.18 | -0.06 | 0.18 |
| Skull length | -0.18 | -0.79 | -0.91 | 0.19 |
| Skull width | 0.28 | -0.08 | 0.25 | -0.29 |
| Skull index | 0.10 | -0.11 | 0.16 | -0.14 |
| Neck length | -0.21 | 0.36 | -0.04 | -0.98 |
| Beak length | -0.30 | 0.01 | 0.30 | -0.16 |
| Beak width | 0.03 | 0.18 | 0.30 | 0.05 |
| Beak index | -0.06 | -0.09 | -0.02 | -0.15 |
| Spur length | 0.05 | -0.08 | 0.67 | 0.54 |
| Wing span | -0.08 | 1.08 | 0.95 | -0.67 |

| Class Means on Canonical Variables | | | | |
|---|---|---|---|---|
| Chicken ecotype | Female chicken ecotypes | | Male chicken ecotypes | |
| | CAN1 | CAN2 | CAN1 | CAN2 |
| Highland | 2.93 | -1.52 | -4.19 | 4.58 |
| Lowland | 0.25 | 1.96 | 5.78 | 0.11 |
| Midland | -2.55 | -1.12 | -3.83 | -3.65 |

(CAN2). Wattle width contributed equally to both canonical variables. CAN1 exhibited higher discriminatory power compared to CAN2, with CAN1 having a comparative advantage of 134% in explaining the morphometric variability among male chicken ecotypes. This suggests that body length, shank length, comb length, earlobe width, earlobe index, wattle width, skull length, skull index, beak length, beak width, spur length, and wingspan are the most important variables for maximizing the separation among male chicken ecotypes.

The canonical discriminant analysis also quantifies the strength of the overall relationships between the linear composites of the predictors (canonical variables) and criterion variables (ecotypes). The significant canonical correlations observed among the female chicken ecotypes

and the first canonical variable (rc = 0.905), as well as the ecotypes and the second canonical variable (rc = 0.849), indicate that the canonical variates account for the variations among the female chicken ecotypes. However, it is worth noting that the first canonical variable exhibited greater discriminative power than the second canonical variable in distinguishing among the female ecotypes.

Similarly, in male chicken ecotypes, the canonical correlation between the male chicken ecotypes and the first canonical variable (rc = 0.979), along with the ecotypes and the second canonical variable (rc = 0.953), demonstrates that the canonical variates explain the differentiation of the male chicken ecotypes. Once again, the first canonical variable displayed more significant discriminatory power than the second canonical variable in discerning the male ecotypes.

The adjusted canonical correlation analysis reveals that the first canonical correlation represents the largest possible multiple correlation among the groups. This correlation is achieved through a linear combination of the quantitative traits. Specifically, the first canonical correlation for male and female chicken ecotypes was 0.979 and 0.905, respectively (Table 6).

In the case of female chicken ecotypes, the first canonical variable, CAN1, can be expressed as a linear combination of the centered variables: CAN1 = -0.30 x body length + -0.17 x body weight + 0.086699 x shank length + -0.19 x comb length + 0.06 x comb width + 0.13 x comb index +. . . -0.08 x wingspan. Additionally, the second canonical correlation (CAN2) can be given as -0.06 x body length + -0.07 x body weight + 0.66 x shank length +—(1.08) x wingspan. These combinations effectively separate the female chickens among the ecotypes. Similarly, for male ecotypes, the first canonical variable (CAN1) can be expressed as CAN1 = 1.18023 x body length + -0.03 x body weight + 0.62 x shank length + 0.77 x comb length + -0.07 x comb width + -0.49 x comb index +. . . + (-0.67) x wingspan. The second canonical variable (CAN2) can be presented as CAN2 = -0.68 x body length + -0.48 x body weight + -0.22 x shank length + -0.09 x comb length + -0.33 x comb width + -0.73 x comb index +. . . + (-0.67) x wingspan. These combinations explain 70.06% and 29.94% of the total morphometric variations among the male chicken ecotypes, effectively differentiating the male chicken ecotypes.

## Cluster analysis

The dendrogram in the cluster analysis reveals the presence of three distinct clusters among the female chicken populations (Fig 3). Moreover, the discriminant analysis also identifies these chicken populations as three distinct clusters, based on the estimated Mahalanobis distance between the three female populations (Table 8). The results of the discriminant function demonstrate that the ecotypes are correctly classified into three separate groups, achieving an overall classification success rate of 97.3% (Table 7). Specifically, the lowland female chicken ecotypes exhibit the highest level of correct classification (98.17%), followed by the midland chicken ecotypes (97.86%), while the highland chicken ecotypes display the lowest level of correct classification (95.37%).

The distances between all pair wise combinations of chicken ecotypes are highly significant (p < 0.001).The largest distance value is observed between highland and midland female chicken ecotypes (30.17),whereas the lowest distance value is recorded between midland and lowland chicken ecotypes (17.35).This suggests a potential gene flow from highland or midland populations to the lowland chicken ecotypes. One possible explanation for this could be the practice of collecting and transporting indigenous chicken products (live chicken and eggs) from both highland and midland regions to the lowland agro-ecology during various occasions, such as Ethiopian religious festivities and the Ethiopian New Year. The price of chicken products in the highland and midland regions is comparatively lower, making it more

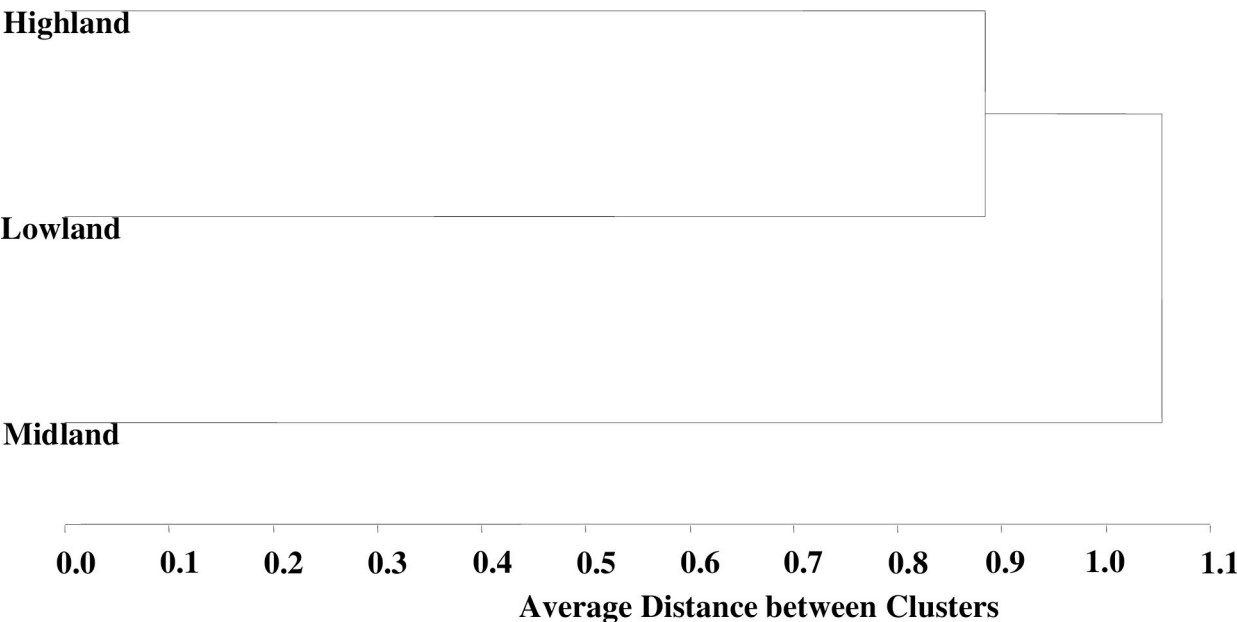

**Fig 3. Dendogram for 412 adult female chicken ecotypes by average LINKAGEcluster.**

affordable for lowland consumers. Additionally, it could be attributed to the seasonal movement of farmers and their domestic animals, including oxen, donkeys, and chickens, from both the highland and midland regions to the lowland agro-ecologies during the rainy season. This migration is driven by the scarcity of cultivated land in both the highland and midland regions.

Similarly, the dendrogram generated from the cluster analysis also reveals that the three male chicken ecotypes are classified into distinct clusters (Table 7 and Fig 4).The discriminant analysis further categorizes the male chicken populations into three separate groups based on the estimated Mahalanobis distance among them (Table 8).The result of the discriminant analysis demonstrates that the male chicken ecotypes are accurately classified into three distinct groups, achieving a total classification success rate of 100% (Table 7).

Furthermore, the analysis confirms that the male chicken ecotypes are correctly classified into three different categories using variations in agro-ecologies as a classification variable, with a perfect classification rate and a 0% error rate for the original male chicken ecotypes. The distance between highland and lowland male chicken ecotypes is 122.09, slightly higher than the distance between lowland and midland chicken ecotypes (105.85), while the smallest distance is observed between highland and midland ecotypes (68.65). All inter-population distances among the male chicken ecotypes are significant. The relatively shorter distance observed between the highland and midland ecotypes suggest a higher level of gene flow between these two chicken ecotypes. One possible explanation for the differences in the

**Table 8. Proximity matrix or Pair wise generalized squared distance to indigenous chicken Ecotypes.**

| From | Female | | | From | Male | | |
|---|---|---|---|---|---|---|---|
| Ecotype | Highland | Lowland | Midland | | Highland | Lowland | Midland |
| Highland | 0 | **19.32** | 30.17 | Highland | 0 | **122.09** | 68.65 |
| Lowland | 19.32 | 0 | **17.35** | Lowland | 122.09 | 0 | **105.85** |
| Midland | 30.17 | 17.35 | 0 | Midland | 68.65 | 105.85 | 0 |

**Table 7. Classification summary for Indigenous chicken ecotypes of Western Tigray.**

| Female chicken ecotypes | | | |
|---|---|---|---|
| From | Highland | Lowland | Midland | Total |
| Highland | **103 (95.37%)** | 5 (4.63%) | 0 | 108(100%) |
| Lowland | 3 (1.83%) | **161 (98.17%)** | 0 | 164 (100%) |
| Midland | 0 | 3 (2.14) | **137 (97.86%)** | 140 (100%) |
| Total | 106 (25.73%) | 169 (41.02%) | 137 (33.25%) | 412 (100%) |
| Priors | 26.2% | 39.8% | 34% | |

= **97.3%** of original female chicken ecotypes are correctly classified with error rates of **2.7%**

| Male chicken ecotypes | | | |
|---|---|---|---|
| From | Highland | Lowland | Midland | Total |
| Highland | **92(100%)** | 0 | 0 | 92(100%) |
| Lowland | 0 | **146(100%)** | 0 | 146(100%) |
| Midland | 0 | 0 | **120(100%)** | 120(100%) |
| Total | 92(25.70%) | 146(40.78%) | 120(33.52%) | 358(100%) |
| priors | 25.70% | 40.78%% | 33.52% | |

= **100%** of original male chicken ecotypes are correctly classified with error rate of **0%**

Diagonal of classification table indicate correctly classified numbers& percentages / each group.

dendrogram order between male and female chicken ecotypes is the lower likelihood of sampling highland chickens from the lowland male chicken population. Since most farmers typically keep only one cock per flock and prioritize more layers for breeding purposes, they often sell or slaughter more male chickens than females.

## Discussion

The average shank length and body weight obtained in the current study are consistent with findings for Ethiopian indigenous chickens [22,35–37]. However, indigenous chickens in southwestern Ethiopia exhibited higher body weight and shorter shank length compared to current result [38].The wingspan and body length mean values in this study were lower than those observed in Sheka zone, Southwest Ethiopia [36], Southwest Ethiopia [38], and Bench Maji zone, Southwest Ethiopia [37]. Additionally, the shank length and wingspan values in

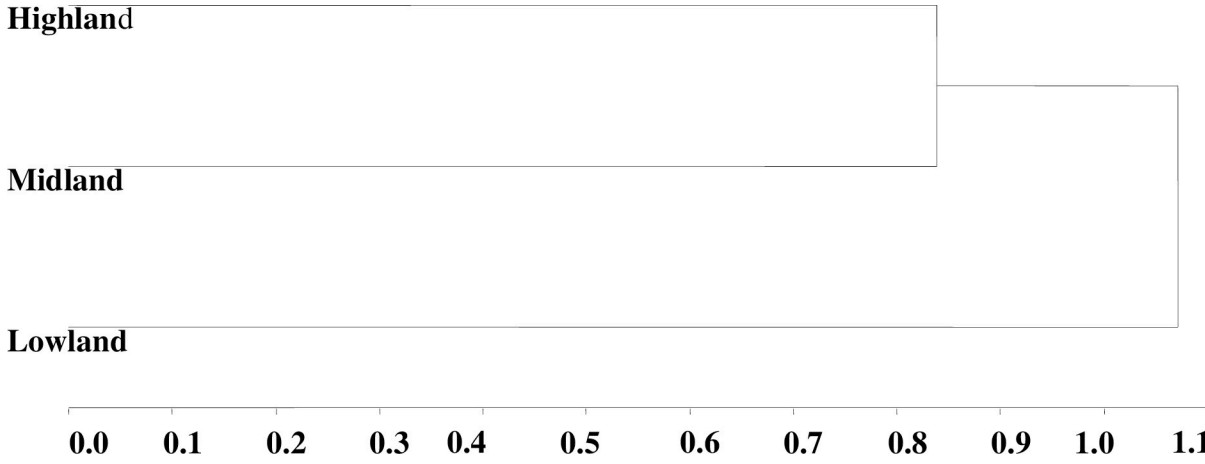

**Fig 4. Dendogram for 358 adult male chicken ecotypes by average LINKAGE cluster.**

Kediri regency, East Java, Indonesia, were higher than those in this study [39]. Tareke [40] found comparable mean values for body length, shank length, wingspan, and beak length, but lower values for body weight, comb length, comb width, and wattle length in indigenous chickens reared in Bale zone, Oromia, Ethiopia. Mearg [41] also reported similar mean values for shank length, but lower mean values for body length and wingspan for indigenous chickens in the central zone of Tigray.The variations in quantitative traits among local chicken populations in different areas of Ethiopia may be attributed to genetic variations, rearing environments, isolation by physical barriers, agroecology, significant geographical distances, human selection pressures, breeding programs, and genetic interactions within the populations.

The significant variations in quantitative traits studied among the three indigenous chicken ecotypes indicate high genetic diversity. Aside from earlobe, skull, and beak indices, mean values for all other traits were significantly higher in cocks than hens. These sexual differences in body measurements align with previous reports from Ethiopia [4,36,42–44]. This can be attributed to sexual dimorphism influenced by growth hormones like androgen and estrogen [44–46].

The PCA analysis revealed that four meaningful PCs explained about 74.26% of the total variations in male chicken ecotypes, while seven PCs explained approximately 68.77% of the total variations in female chicken ecotypes. This finding is consistent with previous reports by Egena et al. [47] and Adedibu et al. [48], who found that the first two retained principal components accounted for the highest variation in Nigerian indigenous chicken populations. Similarly, Adekoya et al. [33] reported that the first three principal components explained 68.201% of the total variations in morphometric traits of Nigerian indigenous chicken types. Udeh and Ogbu [49] found that two to three principal components explained significant variances in different strains of Nigerian chickens. Mearg [41] reported that five principal components accounted for 58.5% of the total variations in morphometric traits of indigenous female chicken populations in Tigray. Yakubu and Ayoade [50] observed two principal components that explained 90.27% of the overall variation in domestic rabbits.

Furthermore, Ukwu et al. [51] found that three principal components accounted for 85.80% of the total variations in 11 egg quality traits of Isa Brown Layer Chickens in Nigeria. Udeh and Ogbu [49] and Egena et al. [47] supported the use of retained principal components as selection criteria for improving body weight or meatiness in broilers and predicting live weight and carcass weight. Yakubu et al. [32] highlighted the importance of principal components in management, conservation, and future selection and breeding programs for Nigerian ducks. Additionally, Udeh [52] demonstrated that principal component-based prediction models are more reliable than interdependent-based models for predicting the body weight of Nigerian rabbits due to the elimination of multicollinearity.

The retained principal components for chicken sex ecotypes can aid in evaluating native chickens for breeding and selection [47,53]. Since there is no association between principal components, selecting animals based on one principal component does not result in a correlated response from other components [54]. Yakubu et al. [46] confirmed that using independent orthogonal indices (principal components) is more appropriate than using original associated linear attributes for predicting chicken body weight, as multicollinearity between interdependent body dimensions can lead to unreliable regression coefficients and inaccurate inferences. In Nigeria, Yakubu et al. [46] found three principal components for normal, naked neck, and frizzled chickens, which were used to predict body weight and select animals with optimal balance. Saikhom et al. [53] reported that two principal components explained 75.7% of the total variation in the morphometric traits of Haringhala Black chickens. Fajemilehin et al. [55] found that principal components explained a significant portion of the variation in the morphological traits of Guinea fowl varieties in Nigeria. The variation in the

number of principal components extracted and their explained variances can result from the differences in the genetic makeup of the ecotypes, environmental factors, or measurement methods. The variation in principal components distinguishing female and male chicken ecotypes in the area may be due to genetic differences linked to body size, reproductive traits, phenotypic variation, sexual dimorphism, and behavioral differences like mating displays or territorial behavior (https://statisticsbyjim.com/basics/principal-component-analysis/).

The stepwise discriminant analysis showed that the following traits, in order of importance, have potential discriminatory power for differentiating the three female chicken ecotypes: earlobe length, wingspan, skull length, shank length, earlobe width, neck length, body length, beak index, beak length, wattle index, body weight, earlobe index, comb index, wattle length, wattle width, and skull index. Similarly, the analysis revealed that the following traits, in order of significance, have a potential discriminating effect for differentiating the three male chicken ecotypes: wingspan, neck length, earlobe length, spur length, body length, skull length, shank length, earlobe index, comb length, wattle length, comb index, beak width, body weight, beak index, wattle index, and wattle width.

This result supports the findings of Mearg [41], who reported that neck length, beak length, body weight, wattle width, wattle length, and height at the back were significant in differentiating chicken populations from different agroecologies. Abdelqader *et al*. [56] found that body weight, body length, heart girth, and back height were highly discriminatory in Jordanian chicken ecotypes. Ogah *et al*. [31] reported that body weight, body width, and body height were important traits for discriminating Nigerian Muscovy duck ecotypes. Adedibu *et al*. [48] found that feather, earlobe, and beak colors, age, and body and neck lengths, accounted for variability in helmeted guinea fowl populations in Nigeria. Eskindir *et al*. [34] found that shank length, body length, comb width, body weight, wingspan, and comb height caused morphological variations in Ethiopian chicken ecotypes. Al-Altiyat [14] found that live weight and carcass weight were important traits for discriminating among chicken populations in Jordan. Yakubu *et al*. [45] also found that foot length, neck length, thigh circumference, and body length were discriminating traits in Nigerian ducks. Al-Altiyat *et al*. [57] reported that comb type and earlobe color differentiated male chicken ecotypes, while beak, earlobe, eye, shank color, and feather distribution were important in discriminating female chickens in Saudi Arabia. Ogah [58] found that body weight, thigh length, and body width were the most important discriminating traits in Nigerian indigenous chicken ecotypes. Daikwo *et al*. [59] found that breast length, body length, shank length, bird height, head circumference, wing length, neck length, and keel length were highly discriminating traits between normal and frizzle-feathered indigenous chickens. Getu *et al*. [60] found that shank length, keel length, wingspan, and beak length were important traits for discriminating indigenous chicken ecotypes in Ethiopia. Adeyemi and Oseni [61] found that body weight, shank length, and abdomen circumference were discriminatory traits in Nigerian indigenous turkeys. Male chickens in the Hadiya zone of the Southern Regional State of Ethiopia were primarily distinguished by their shank length and wingspan, while female chickens were mainly differentiated by their wingspan, body weight, chest circumference, and body length [23]. Melesse *et al*. [21] similarly identified body length and wingspan as distinguishing traits of indigenous chicken populations in the Sheka, Keffa, Bale, and Metekel zones of Ethiopia.

The canonical discriminant analysis yielded two canonical variables for females and two for males, accounting for 100% of the total variations. The first canonical variable explained 63.58% of the variations in females and 70.06% in males, while the second canonical variable accounted for 36.42% in females and 29.94% in males. This aligns with similar studies conducted in Ethiopia [60] and Nigeria [58]. This approach simplifies animal evaluation by

reducing the required number of variables and overcoming the challenge of assigning appropriate weights to each original trait when creating a general index [17].

Canonical loading measures the simple linear correlation between the original independent variables and the dependent canonical variables. It indicates the variance shared by the observed variables with the canonical variate and represents their relative contribution to each canonical variate function [62].

For female chicken ecotypes, the following traits had higher weights in extracting CAN1: body length, body weight, earlobe length, earlobe width, earlobe index, skull width, and beak length. The traits shank length, comb length, comb width, comb index, wattle length, wattle width, wattle index, skull length, skull index, neck length, beak width, beak index, spur length, and wingspan significantly contributed to the second canonical variable, CAN2.

In male chicken ecotypes, the traits with higher weights in extracting CAN1 were body length, shank length, comb length, earlobe width, earlobe index, wattle width, skull length, skull index, beak length, beak width, spur length, and wingspan. The traits body weight, comb width, comb index, earlobe length, wattle length, wattle index, skull width, neck length, and beak index significantly contributed to the second canonical variable, CAN2. Similar results have been reported in chicken populations from other regions in Ethiopia [42,44,61]. In Nigeria, Ogah [58] found that the two discriminant functions accounted for 100% of the total variations among Nigerian indigenous chicken ecotypes.

The dendrogram in cluster analysis revealed three distinct clusters formed by both sexes of chicken ecotypes. Discriminant analysis further supported these clusters by calculating the Mahalanobis distance between the populations. For female chicken ecotypes, the discriminant function correctly classified them into three groups, achieving a classification success rate of 97.3% with an error rate of 2.7%. Among the female ecotypes, the lowland chicken ecotypes had the highest classification accuracy (98.175%), followed by the midland ecotypes (97.86%), while the highland ecotypes had the lowest classification accuracy (95.375%).

The results further confirmed the accurate classification of male chicken ecotypes into three distinct categories based on agro-ecologies. All original male chicken ecotypes were correctly classified, resulting in a 0% error rate. This finding supports the importance of considering genetic, ecological, morphological, and productive aspects in population classification, as emphasized by Yunon et al. [63]. However, it contradicts the findings of Turan et al. [64], who suggested that achieving a 100% accurate classification of animals to their original population based on phenotypic measurements is challenging.Muluneh et al. [25] also reported contrasting results, indicating that the overall classification rates for the female and male chicken populations were 57.47% and 69.97%, respectively, in the three zones of the Amhara National Regional State of Ethiopia.

The Mahalanobis distance analysis revealed clear differentiation among chicken populations of both sexes. The highest distance was observed between highland and midland female ecotypes (30.17), while the lowest distance was found between midland and lowland ecotypes (17.35). This indicates a possible gene flow from the highland or midland ecotypes to the lowland ecotypes. This could be due to the practice of collecting and transporting indigenous chicken products (live chicken and eggs) from the highlands and midlands to the lowland agroecology. This is often done by local chicken producers or traders, especially during Ethiopian religious festivals and the Ethiopian New Year. The reason behind this is that the prices of chicken products are significantly lower in the highlands and midlands compared to the lowland agroecology. Additionally, seasonal movement of farmers with their animals, including oxen, donkeys, and chickens, from both agroecologies to the lowlands during the rainy season could be another contributing factor. This movement allows them to cultivate their own or rented land, as there is a limited availability of cultivated land in the highland and midland agroecologies.

The highest distance value was observed between highland and lowland male chicken ecotypes (122.09), while the lowest distance value was found between midland and lowland male chicken ecotypes (68.65). This suggests a relatively high gene flow between the lowland and midland ecotypes.

The likely reason for the difference in the order (distance) of chicken ecotypes in the dendrogram between males and females may be due to the lower likelihood of sampling highland male chickens from a lowland male chicken population. This is because farmers usually keep only one cock per flock and have more hens for breeding. Consequently, they tend to sell or slaughter more males than females.

The overall percentage of correctly classified individual chickens was 97.3% in females and 100% in males, which is higher than the reported 92.5% for Nigerian Muscovy ducks [31]. Adekoya *et al.* [33] also found lower overall correct classification figures for Nigerian chicken genotypes, with 56% for three clusters and percentages for specific types ranging from 78.6% to 60%. Similarly, Gwaza *et al.* [65] reported an overall proportion of 37.72% for correctly classified Nigerian chicken ecotypes in Guinea Savannah, with percentages for specific groups ranging from 42.8% to 59.4%.Lowercorrect classification percentages were also reported for indigenous chicken populations in the Sheka, Keffa, Bale, and Metekel zones of Ethiopia [21], as well as in the Hadiya zone of the Southern Regional State of Ethiopia [23].

The distances observed in both chicken ecotype sexes in this study were significantly higher than those reported in previous research on the morphometric variation evaluation of Nigerian Muscovy duck ecotypes [32]. Low distances were also found in previous studies on Nigerian chicken genotypes [58]. Similarly, the distances obtained among all female chicken ecotypes were lower than those found in previous research on Muscovy duck ecotypes [31]. However, the distance between male ecotypes was greater than that reported in previous research on Muscovy ducks [31]. Additionally, small distances were reported between chicken ecotypes in Saudi Arabia [57],the central zone of Tigray [20], and three zones of the Amhara National regional state of Ethiopia [25]. These findings contrast with the research on indigenous, commercial layer, and broiler chickens in Jordan [14].

## Conclusion

Significant quantitative morphological variations were observed among the three indigenous chicken ecotypes. Four and seven principal components were extracted for male and female chicken ecotypes, respectively, and they explained a significant portion of the total variability in morphometric trait. Two significant canonical variables explained a significant portion of morphometric trait variability in male and female ecotypes. Wingspan, neck length, earlobe length, spur length, body length, and skull length were important traits for discriminating male ecotypes, while earlobe length, wingspan, skull length, and shank length were the most significant traits for distinguishing female ecotypes. The discriminant analysis demonstrated a high classification success rate of 97.3% for females and 100% for males, with low error rates. Cluster analysis further classified the indigenous chicken flocks into Lowland, Midland, and Highland ecotypes, highlighting their genetic diversity. DNA studies are recommended to validate the observed morphological variations among indigenous chicken ecotypes, supporting conservation efforts and facilitating genetic improvement.

## Supporting information

**S1 Table. Mean, minimum, and maximum values for quantitative traits of three female chicken ecotypes.**
(DOCX)

**S2 Table. Mean, minimum, and maximum values for quantitative traits of three male chicken ecotypes.**
(DOCX)

## Acknowledgments

The authors are grateful to the Tigray Agricultural Research Institute's Humera Agricultural Research Center for facilitating transportation. Our heartfelt thanks and gratitude go to those who contributed directly and indirectly to the project's execution.

## Author Contributions

**Conceptualization:** Shishay Markos, Berhanu Belay, Tadelle Dessie.

**Data curation:** Shishay Markos.

**Formal analysis:** Shishay Markos.

**Funding acquisition:** Shishay Markos.

**Investigation:** Shishay Markos, Berhanu Belay, Tadelle Dessie.

**Methodology:** Shishay Markos, Berhanu Belay, Tadelle Dessie.

**Project administration:** Shishay Markos, Berhanu Belay.

**Software:** Shishay Markos, Berhanu Belay.

**Supervision:** Berhanu Belay, Tadelle Dessie.

**Validation:** Shishay Markos, Berhanu Belay, Tadelle Dessie.

**Visualization:** Shishay Markos, Berhanu Belay, Tadelle Dessie.

**Writing – original draft:** Shishay Markos.

**Writing – review & editing:** Shishay Markos, Berhanu Belay, Tadelle Dessie.

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
