## [Decision Letter · Decision Letter 0]

24 May 2023

PONE-D-23-12190Morphometric Differentiation of Three Chicken Ecotypes of Ethiopia Using Multivariate AnalysisPLOS ONE

Dear Dr. Markos,

Thank you for submitting your manuscript to PLOS ONE. After careful consideration, we feel that it has merit but does not fully meet PLOS ONE’s publication criteria as it currently stands. Therefore, we invite you to submit a revised version of the manuscript that addresses the points raised during the review process.

Dear Dr., Shishay Markos

Thank you for submitting your manuscript to PLOS ONE. After careful consideration, we feel that it has merit but does not fully meet PLOS ONE’s publication criteria as it currently stands. Therefore, we invite you to submit a revised version of the manuscript that addresses the points raised during the review process.

Thank you for submitting your manuscript to PLOS ONE. After careful consideration, we have decided that your manuscript needs Minor Revision.

Kind regards,

Prof. Lamiaa Mostafa Radwan, Ph.D.

Academic Editor

PLOS ONE

Reviewer1

What is the standard to use almost 1:1 ratio of male and female (146 male and 164 females) chicken in your study

*You already have fixed the chickens in to their perspective agro-ecologies; so, why you need to conduct cluster analysis?

*I do not think some traits to be included as discriminating traits; for example wattle length and earlobe length in stepwise selection of traits (Table 5).

*Some inconsistencies…. Please use either” local” or “indigenous” for the chicken ecotypes you used for the study. My recommendation is to say “indigenous” rather than saying “local” along the manuscript

Reviewer 2

The article Morphometric Differentiation of Three Chicken Ecotypes of Ethiopia Using Multivariate Analysis is of interest. The authors have measured 21 morphometric traits of 770 chickens from three agro-ecologies of Tigray. Then Multivariate analysis was done and they found that among different traits earlobe length, Wingspan, skull length, and shank length were the most important traits for discriminating among female chicken ecotypes and wingspan, neck length, earlobe length, spur length, body length, and shank length were the most important discriminatory traits among male chicken ecotypes.

I think that the following areas should be dealt with in revision of this paper:

1. Introduction: Ethiopia is thought to have the largest livestock population…………………million beehives. The reference for that data is of 2016. Please include the data from recent references.

2. ‘Poor understanding of the production system and absence of comprehensive breeding strategies Efforts to boost the performance……..’ write the sentence properly

3. ‘Baseline information on…………………….utilization and improvements’. Split the sentence and make simple sentences for easy understanding.

4. ‘Rosario et al. have stated that the processesinvolved in the regulation of morphological traits in chickens are too complex to explain only in the univariate analysis as associated traits are biologically correlated owing to the pleiotropic effect of genes and loci linkages’. Split the sentence and make simple sentences for easy understanding.

5. ‘Multivariate statistical techniques are suitable approaches to analyzing……’ replace ‘to’ with ‘for’

6. ‘Thus, this study was designed to assess the genetic diversity and differentiation of the three local ecotypes of the western zone of the Tigray regional state of Ethiopia by using multivariate analysis and considering morphometric traits’. In this study only some morphometric traits were measured so how the genetic diversity was estimated in the study?

7. The materials and methods section needs more detail. This is especially important in the light of strong efforts on the part of the scientific community to improve replicability.

8. The Results section needs clarification and tighter organization.

9. The discussion section needs to be tightened up with respect to writing. It is overly long and quite speculative.

10 Conclusion section is also lengthy reduce the section and keep only the salient findings.

We look forward to receiving your revised manuscript.

Kind regards,

Lamiaa Mostafa Radwan, Ph.D.

Academic Editor

PLOS ONE

Journal Requirements:

"The authors are grateful to the Humera Agricultural Research Centre of the Tigray Agricultural Research Institute for funding the project with grant number 2130207. Our deepest appreciation and heartfelt thanks go to those individuals, including all sample Kebele administrations, farmers (chicken owners), Kafta Humera, Welkait, and Tsegede wereda agricultural offices, experts, and development agents that were involved directly and indirectly in the execution of the project."

"Funding: The author: Shishay Markos received funding through grant number: 2130207 from Humera Agricultural Research Centre of the Tigray Agricultural Research Institute 

 (https://www.asti.cgiar.org/ethiopia/directory/tigray-agricultural-research-institute-tari). 

 The funder had no role in study design, data collection and analysis, decision to publish, or

preparation of the manuscript."

7. We note that Figure 1] in your submission contain [map/satellite] images which may be copyrighted. All PLOS content is published under the Creative Commons Attribution License (CC BY 4.0), which means that the manuscript, images, and Supporting Information files will be freely available online, and any third party is permitted to access, download, copy, distribute, and use these materials in any way, even commercially, with proper attribution. For these reasons, we cannot publish previously copyrighted maps or satellite images created using proprietary data, such as Google software (Google Maps, Street View, and Earth). For more information, see our copyright guidelines: http://journals.plos.org/plosone/s/licenses-and-copyright.

Reviewers' comments:

Reviewer's Responses to Questions

**Comments to the Author**

1. Is the manuscript technically sound, and do the data support the conclusions?

Reviewer #1: Yes

Reviewer #2: Partly

2. Has the statistical analysis been performed appropriately and rigorously? 

Reviewer #1: Yes

Reviewer #2: Yes

3. Have the authors made all data underlying the findings in their manuscript fully available?

Reviewer #1: Yes

Reviewer #2: No

4. Is the manuscript presented in an intelligible fashion and written in standard English?

Reviewer #1: Yes

Reviewer #2: No

5. Review Comments to the Author

Reviewer #1: *What is the standard to use almost 1:1 ratio of male and female (146 male and 164 females) chicken in your study

*You already have fixed the chickens in to their perspective agro-ecologies; so, why you need to conduct cluster analysis?

*I do not think some traits to be included as discriminating traits; for example wattle length and earlobe length in stepwise selection of traits (Table 5).

*Some inconsistencies…. Please use either” local” or “indigenous” for the chicken ecotypes you used for the study. My recommendation is to say “indigenous” rather than saying “local” along the manuscript

Reviewer #2: The article Morphometric Differentiation of Three Chicken Ecotypes of Ethiopia Using Multivariate Analysis is of interest. The authors have measured 21 morphometric traits of 770 chickens from three agro-ecologies of Tigray. Then Multivariate analysis was done and they found that among different traits earlobe length, Wingspan, skull length, and shank length were the most important traits for discriminating among female chicken ecotypes and wingspan, neck length, earlobe length, spur length, body length, and shank length were the most important discriminatory traits among male chicken ecotypes.

I think that the following areas should be dealt with in revision of this paper:

1. Introduction: Ethiopia is thought to have the largest livestock population…………………million beehives. The reference for that data is of 2016. Please include the data from recent references.

2. ‘Poor understanding of the production system and absence of comprehensive breeding strategies Efforts to boost the performance……..’ write the sentence properly

3. ‘Baseline information on…………………….utilization and improvements’. Split the sentence and make simple sentences for easy understanding.

4. ‘Rosario et al. have stated that the processesinvolved in the regulation of morphological traits in chickens are too complex to explain only in the univariate analysis as associated traits are biologically correlated owing to the pleiotropic effect of genes and loci linkages’. Split the sentence and make simple sentences for easy understanding.

5. ‘Multivariate statistical techniques are suitable approaches to analyzing……’ replace ‘to’ with ‘for’

6. ‘Thus, this study was designed to assess the genetic diversity and differentiation of the three local ecotypes of the western zone of the Tigray regional state of Ethiopia by using multivariate analysis and considering morphometric traits’. In this study only some morphometric traits were measured so how the genetic diversity was estimated in the study?

7. The materials and methods section needs more detail. This is especially important in the light of strong efforts on the part of the scientific community to improve replicability.

8. The Results section needs clarification and tighter organization.

9. The discussion section needs to be tightened up with respect to writing. It is overly long and quite speculative.

10 Conclusion section is also lengthy reduce the section and keep only the salient findings.

6. PLOS authors have the option to publish the peer review history of their article (what does this mean?). If published, this will include your full peer review and any attached files.

Reviewer #1: No

Reviewer #2: No

---

## [Author Response · Author response to Decision Letter 0]

20 Jun 2023

We provided responses to reviewers' and editors' comments point by point, and we submitted as response to reviewers

---

## [Decision Letter · Decision Letter 1]

19 Jul 2023

PONE-D-23-12190R1Morphometric Differentiation of Three Chicken Ecotypes of Ethiopia Using Multivariate AnalysisPLOS ONE

Dear Dr. Markos,

Thank you for submitting your manuscript to PLOS ONE. After careful consideration, we feel that it has merit but does not fully meet PLOS ONE’s publication criteria as it currently stands. Therefore, we invite you to submit a revised version of the manuscript that addresses the points raised during the review process.

Dear Dr., Shishay Markos, MSc

Thank you for submitting your manuscript to PLOS ONE. After careful consideration, we have decided that your manuscript needs Minor Revision.

Kind regards,

Comment Editor

The study provides valuable scientific information on biometrical measurements of chicken.

However, the manuscript needs to make some important modifications

Need a linguistic reformulation of the English language I recommend sending it to the authorities editing languageThe difference of ages affects the measured data, so the effect of age must be included in the statistical model used to analyze the data

Prof. Lamiaa Mostafa Radwan, Ph.D.

Academic Editor

PLOS ONE

**Reviewer1**

I have satisfied;

But for an other time while working such kind of works, it is better to balance the number of male and female animals following the guideline for phenotypic characterization of animals

**Reviewer2**

The study presents interesting morphometric insights on the three chicken ecotypes of Ethiopia using multivariate analysis. However, there are certain queries which need to be properly addressed. Moreover, English language needs to be thoroughly edited and improved. Use of unscientific words and unnecessarily long sentences has to be avoided and the overall content needs to be made more concise and clear

Methods

• Sampling technique: 6 months old or older chicken seems to be a vague criteria especially in the light of the fact that inclusion of older chickens may influence the individual trait measurements and may hide the true picture of the traits in the three ecotypes. Age should be considered as a separate factor for more accuracy in results

• Selection of farmers based on possession of 3 or more chickens: Three numbers of chickens is too less to provide the accurate description of morphometric traits and hence, characterize a population

• Please describe the visible appearance of the chickens ecotypes

• The same temperature in the study area of both Weynadega (midland) and Dega (highland) seems confusing as different ecological regions should have different temperatures and other weather conditions

• Population size for different ecotypes is less so as to warrant enough accuracy

Results

• PCA plots discriminating the three ecotype populations would have greatly helped in easier interpretation. This will also help to check for overlaps in different populations for further clarity

• PCA and step wise discriminant analysis provided enough clarity regarding the discrimination of the populations and the most important traits for discriminating the three populations. Was there some special reason as to why Canonical discriminant analysis was done?

• 7 PCs for discriminating female chicken ecotypes does not seems to make much sense…Comment

We look forward to receiving your revised manuscript.

Kind regards,

Lamiaa Mostafa Radwan, Ph.D.

Academic Editor

PLOS ONE

Journal Requirements:

Additional Editor Comments:

Dear Dr., Shishay Markos, MSc

Thank you for submitting your manuscript to PLOS ONE. After careful consideration, we have decided that your manuscript needs Minor Revision.

Kind regards,

Comment Editor

The study provides valuable scientific information on biometrical measurements of chicken.

However, the manuscript needs to make some important modifications

1- Need a linguistic reformulation of the English language I recommend sending it to the authorities editing language

2- The difference of ages affects the measured data, so the effect of age must be included in the statistical model used to analyze the data

Prof. Lamiaa Mostafa Radwan, Ph.D.

Academic Editor

PLOS ONE

Reviewer1

I have satisfied;

But for an other time while working such kind of works, it is better to balance the number of male and female animals following the guideline for phenotypic characterization of animals

Reviewer2

The study presents interesting morphometric insights on the three chicken ecotypes of Ethiopia using multivariate analysis. However, there are certain queries which need to be properly addressed. Moreover, English language needs to be thoroughly edited and improved. Use of unscientific words and unnecessarily long sentences has to be avoided and the overall content needs to be made more concise and clear

Methods

• Sampling technique: 6 months old or older chicken seems to be a vague criteria especially in the light of the fact that inclusion of older chickens may influence the individual trait measurements and may hide the true picture of the traits in the three ecotypes. Age should be considered as a separate factor for more accuracy in results

• Selection of farmers based on possession of 3 or more chickens: Three numbers of chickens is too less to provide the accurate description of morphometric traits and hence, characterize a population

• Please describe the visible appearance of the chickens ecotypes

• The same temperature in the study area of both Weynadega (midland) and Dega (highland) seems confusing as different ecological regions should have different temperatures and other weather conditions

• Population size for different ecotypes is less so as to warrant enough accuracy

Results

• PCA plots discriminating the three ecotype populations would have greatly helped in easier interpretation. This will also help to check for overlaps in different populations for further clarity

• PCA and step wise discriminant analysis provided enough clarity regarding the discrimination of the populations and the most important traits for discriminating the three populations. Was there some special reason as to why Canonical discriminant analysis was done?

• 7 PCs for discriminating female chicken ecotypes does not seems to make much sense…Comment

Reviewers' comments:

Reviewer's Responses to Questions

**Comments to the Author**

1. If the authors have adequately addressed your comments raised in a previous round of review and you feel that this manuscript is now acceptable for publication, you may indicate that here to bypass the “Comments to the Author” section, enter your conflict of interest statement in the “Confidential to Editor” section, and submit your "Accept" recommendation.

Reviewer #1: All comments have been addressed

Reviewer #3: (No Response)

2. Is the manuscript technically sound, and do the data support the conclusions?

Reviewer #1: Yes

Reviewer #3: Yes

3. Has the statistical analysis been performed appropriately and rigorously? 

Reviewer #1: Yes

Reviewer #3: Yes

4. Have the authors made all data underlying the findings in their manuscript fully available?

Reviewer #1: Yes

Reviewer #3: Yes

5. Is the manuscript presented in an intelligible fashion and written in standard English?

Reviewer #1: Yes

Reviewer #3: Yes

6. Review Comments to the Author

Reviewer #1: I have satisfied;

But for an other time while working such kind of works, it is better to balance the number of male and female animals following the guideline for phenotypic characterization of animals

Reviewer #3: The study presents interesting morphometric insights on the three chicken ecotypes of Ethiopia using multivariate analysis. However, there are certain queries which need to be properly addressed. Moreover, English language needs to be thoroughly edited and improved. Use of unscientific words and unnecessarily long sentences has to be avoided and the overall content needs to be made more concise and clear

Methods

• Sampling technique: 6 months old or older chicken seems to be a vague criteria especially in the light of the fact that inclusion of older chickens may influence the individual trait measurements and may hide the true picture of the traits in the three ecotypes. Age should be considered as a separate factor for more accuracy in results

• Selection of farmers based on possession of 3 or more chickens: Three numbers of chickens is too less to provide the accurate description of morphometric traits and hence, characterize a population

• Please describe the visible appearance of the chickens ecotypes

• The same temperature in the study area of both Weynadega (midland) and Dega (highland) seems confusing as different ecological regions should have different temperatures and other weather conditions

• Population size for different ecotypes is less so as to warrant enough accuracy

Results

• PCA plots discriminating the three ecotype populations would have greatly helped in easier interpretation. This will also help to check for overlaps in different populations for further clarity

• PCA and step wise discriminant analysis provided enough clarity regarding the discrimination of the populations and the most important traits for discriminating the three populations. Was there some special reason as to why Canonical discriminant analysis was done?

• 7 PCs for discriminating female chicken ecotypes does not seems to make much sense…Comment

7. PLOS authors have the option to publish the peer review history of their article (what does this mean?). If published, this will include your full peer review and any attached files.

Reviewer #1: No

Reviewer #3: No

---

## [Author Response · Author response to Decision Letter 1]

23 Aug 2023

We gave responses to each reviewer's and editor's comments point by point, and we incorporated them in our response to the reviewers. Finally, we submitted our revised version.

---

## [Decision Letter · Decision Letter 2]

18 Sep 2023

PONE-D-23-12190R2Morphometric Differentiation of Three Chicken Ecotypes of Ethiopia Using Multivariate AnalysisPLOS ONE

Dear Dr. Markos,

Thank you for submitting your manuscript to PLOS ONE. After careful consideration, we feel that it has merit but does not fully meet PLOS ONE’s publication criteria as it currently stands. Therefore, we invite you to submit a revised version of the manuscript that addresses the points raised during the review process.

Dear Dr., Shishay Markos, MSc

Thank you for submitting your manuscript to PLOS ONE. After careful consideration, we have decided that your manuscript needs Minor Revision.

Kind regards,

Prof. Lamiaa Mostafa Radwan, Ph.D.

Academic Editor

PLOS ONE

Editor Comments:

1- The conclusion section needs to be refined before publication because there are many repeats with the results section.

2- English language of this paper needs checking.

**Reviewer1**

The article entitled” Morphometric Differentiation of Three Chicken Ecotypes of Ethiopia Using Multivariate Analysis” has scientific merit and useful information. The authors have addressed the queries raised by me during the Review process. The article may be accepted for publication in PLOS one

.

**Reviewer2**

The conclusion section need to be refined before publication, because there are many repeats with the results section

We look forward to receiving your revised manuscript.

Kind regards,

Lamiaa Mostafa Radwan, Ph.D.

Academic Editor

PLOS ONE

Journal Requirements:

Additional Editor Comments:

Dear Dr., Shishay Markos, MSc

Thank you for submitting your manuscript to PLOS ONE. After careful consideration, we have decided that your manuscript needs Minor Revision.

Kind regards,

Prof. Lamiaa Mostafa Radwan, Ph.D.

Academic Editor

PLOS ONE

Editor Comments:

1- The conclusion section needs to be refined before publication because there are many repeats with the results section.

2- English language of this paper needs checking.

Reviewer1

The article entitled” Morphometric Differentiation of Three Chicken Ecotypes of Ethiopia Using Multivariate Analysis” has scientific merit and useful information. The authors have addressed the queries raised by me during the Review process. The article may be accepted for publication in PLOS one

.

Reviewer2

The conclusion section need to be refined before publication, because there are many repeats with the results section

Reviewers' comments:

Reviewer's Responses to Questions

**Comments to the Author**

1. If the authors have adequately addressed your comments raised in a previous round of review and you feel that this manuscript is now acceptable for publication, you may indicate that here to bypass the “Comments to the Author” section, enter your conflict of interest statement in the “Confidential to Editor” section, and submit your "Accept" recommendation.

Reviewer #3: All comments have been addressed

Reviewer #4: All comments have been addressed

2. Is the manuscript technically sound, and do the data support the conclusions?

Reviewer #3: Partly

Reviewer #4: Yes

3. Has the statistical analysis been performed appropriately and rigorously? 

Reviewer #3: Yes

Reviewer #4: Yes

4. Have the authors made all data underlying the findings in their manuscript fully available?

Reviewer #3: (No Response)

Reviewer #4: Yes

5. Is the manuscript presented in an intelligible fashion and written in standard English?

Reviewer #3: Yes

Reviewer #4: Yes

6. Review Comments to the Author

Reviewer #3: The article entitled” Morphometric Differentiation of Three Chicken Ecotypes of Ethiopia Using Multivariate Analysis” has scientific merit and useful information. The authors have addressed the queries raised by me during the Review process. The article may be accepted for publication in PLOS one

(U.Rajkumar)

Principal Scientist & Head

Poultry Genetics & Breeding

Reviewer #4: The conclusion section need to be refined before publication, because there are many repeats with the results section.

7. PLOS authors have the option to publish the peer review history of their article (what does this mean?). If published, this will include your full peer review and any attached files.

Reviewer #3: No

Reviewer #4: No

---

## [Author Response · Author response to Decision Letter 2]

7 Nov 2023

We responded to each editor's and reviewer's comments point by point, incorporating them into the manuscript. We then submitted these responses as suggested to the reviewers.

---

## [Decision Letter · Decision Letter 3]

6 Dec 2023

PONE-D-23-12190R3Morphometric Differentiation of Three Chicken Ecotypes of Ethiopia Using Multivariate AnalysisPLOS ONE

Dear Dr. Markos,

Thank you for submitting your manuscript to PLOS ONE. After careful consideration, we feel that it has merit but does not fully meet PLOS ONE’s publication criteria as it currently stands. Therefore, we invite you to submit a revised version of the manuscript that addresses the points raised during the review process.

We look forward to receiving your revised manuscript.

Kind regards,

Miquel Vall-llosera Camps

Senior Editor

PLOS ONE

on behalf of

Lamiaa Mostafa Radwan, Ph.D.

Academic Editor

PLOS ONE

Request from the Editorial Staff:

During our final internal checks on this submission, we noticed that this manuscript is very closely related to the following papers, of which you are an author:

Morpho-biometric characterization of indigenous chicken ecotypes in North-western Ethiopia", published in PLOS ONE (https://doi.org/10.1371/journal.pone.0286299)

As outlined in our criteria if related work has been published elsewhere, authors must describe its relation to the submitted work (http://journals.plos.org/plosone/s/criteria-for-publication#loc-2). For more information about our policy on related manuscripts please see http://journals.plos.org/plosone/s/ethical-publishing-practice#loc-submission-and-publication-of-related-studies.

Before we proceed further with your PLOS ONE submission, we ask that you make reference of pone.0286299 in your manuscript and contextualise the study relative this previous work. Please clarify the differences in the research questions addressed and the data sets used in these related manuscripts, explain how the work described in your PLOS ONE submission advances on that described in this related paper.

We sincerely apologise that we did not notice this issue sooner in the review process, but hope you understand the reason for this decision. We are monitoring your submission closely and please rest assured that we are doing everything we can to avoid any further delays. 

Thank you for your time and attention.

Reviewers' comments:

Reviewer's Responses to Questions

**Comments to the Author**

1. If the authors have adequately addressed your comments raised in a previous round of review and you feel that this manuscript is now acceptable for publication, you may indicate that here to bypass the “Comments to the Author” section, enter your conflict of interest statement in the “Confidential to Editor” section, and submit your "Accept" recommendation.

Reviewer #4: All comments have been addressed

2. Is the manuscript technically sound, and do the data support the conclusions?

Reviewer #4: Yes

3. Has the statistical analysis been performed appropriately and rigorously? 

Reviewer #4: Yes

4. Have the authors made all data underlying the findings in their manuscript fully available?

Reviewer #4: Yes

5. Is the manuscript presented in an intelligible fashion and written in standard English?

Reviewer #4: Yes

6. Review Comments to the Author

Reviewer #4: Modify all complete and receivable, Dual publication, research ethics or publication ethics are all in accordance with requirements and regulations

7. PLOS authors have the option to publish the peer review history of their article (what does this mean?). If published, this will include your full peer review and any attached files.

Reviewer #4: No

---

## [Author Response · Author response to Decision Letter 3]

10 Dec 2023

We responded to each reviewer's and editor's comments point by point, and we submitted it as a separate response to the reviewers.

---

## [Editor Report · Decision Letter 4]

4 Jan 2024

PONE-D-23-12190R4Morphometric Differentiation of Three Chicken Ecotypes of Ethiopia Using Multivariate AnalysisPLOS ONE

Dear Dr. Markos,

Thank you for submitting your manuscript to PLOS ONE. After careful consideration, we feel that it has merit but does not fully meet PLOS ONE’s publication criteria as it currently stands. Therefore, we invite you to submit a revised version of the manuscript that addresses the points raised during the review process.

Thank you to the author for the response to clarify the differences between the previous and current manuscript, but your responses are only for the auditors

This was not clear in the introduction and discussed results of the manuscript. Please clarify the differences between the previous and current research in the text of writing the manuscript, 

please write different color marks highlighted in the paragraph and all that is added in this is learned on the highlighter to facilitate its review

We look forward to receiving your revised manuscript.

Kind regards,

Lamiaa Mostafa Radwan, Ph.D.

Academic Editor

PLOS ONE

Journal Requirements:

Additional Editor Comments:

Thank you to the author for the response to clarify the differences between the previous and current manuscript, but your responses are only for the auditors

This was not clear in the introduction and discussed results of the manuscript. Please clarify the differences between the previous and current research in the text of writing the manuscript,

please write different color marks highlighted in the paragraph and all that is added in this is learned on the highlighter to facilitate its review

---

## [Author Response · Author response to Decision Letter 4]

27 Jan 2024

We give responses to each reviewer's and editors' comments point by point and we submit it as response to reviewers.

---

## [Editor Report · Decision Letter 5]

1 Feb 2024

Morphometric Differentiation of Three Chicken Ecotypes of Ethiopia Using Multivariate Analysis

PONE-D-23-12190R5

Dear Dr. Markos,

We’re pleased to inform you that your manuscript has been judged scientifically suitable for publication and will be formally accepted for publication once it meets all outstanding technical requirements.

Kind regards,

Lamiaa Mostafa Radwan, Ph.D.

Academic Editor

PLOS ONE

Additional Editor Comments (optional):

Accept
---

## [Editor Report · Acceptance letter]

19 Feb 2024

PONE-D-23-12190R5 

PLOS ONE

Dear Dr. Markos, 

I'm pleased to inform you that your manuscript has been deemed suitable for publication in PLOS ONE. Congratulations! Your manuscript is now being handed over to our production team.

Kind regards, 

on behalf of

Prof. Dr. Lamiaa Mostafa Radwan 

Academic Editor

PLOS ONE